# Distribution and Speciation of Rare Earth Elements in Coal Fly Ash from the Qianxi Power Plant, Guizhou Province, Southwest China

**Lun Wu** [ID], **Liqiang Ma, Gen Huang, Jihui Li** *[ID] **and Hongxiang Xu**

School of Chemical and Environmental Engineering, China University of Mining and Technology-Beijing, Beijing 100083, China
* Correspondence: lijihui@cumtb.edu.cn; Tel.: +86-152-0131-3515

**Abstract:** Coal fly ash (CFA), hazardous to the environment and human health, has been considered to be a potential alternative source for rare earth elements (REEs) in recent years. However, information on how REEs distribute and occur in coal fly ash is still incomplete. In this work, particle size analysis, inductively coupled plasma–mass spectrometry, scanning electron microscopy, and electron probe microanalysis were applied to study the occurrence and distribution of REEs in a fly ash sample from the Qianxi coal-fired power plant in Guizhou province. The results show that the REEs content in the CFA was 630.51 ppm. Wet grinding-enhanced leaching experiments revealed that a part of the rare earth particles was encapsulated within the glass body. These rare earth particles could be liberated and released to a certain extent by wet grinding, which would increase the acid-leaching recovery of REEs from 23.49% to 41.68%. This study classifies the speciation of REEs in coal fly ash as (1) amorphous glassy particles with REE minerals or compounds encapsulated inside; (2) amorphous glassy particles with REEs distributed throughout; and (3) discrete REE minerals or compounds. The results of this study are a basis for developing an economically viable and environmentally sustainable technology for recovering REEs from CFA.

**Keywords:** coal fly ash; rare earth elements; SEM-EDS; particle size; grinding; recovery

## 1. Introduction

According to statistics, China has been consuming approximately 4 billion tons of coal annually in recent years, more than half of which commercial power plants use to generate electricity [1–4]. The coal fly ash (CFA) produced by coal combustion in coal-fired power plants is a typical industrial solid waste that has led to severe environmental issues in China. Usually, in China's coal-fired power plants, every four tons of coal can produce one ton of coal ash. It is conservatively estimated that the annual output of coal ash in China has reached 500–550 million tons [5,6]. If not adequately managed, large-scale fly ash emissions will seriously threaten China's environment and public health [7]. During the 1970s, researchers and scientists began investigating the appropriate techniques for reusing CFA to prevent contamination of the environment [8,9], and until today coal fly ash is mainly reused in the construction industry. Besides studies investigating better methods of reusing CFA in the construction industry as materials in the production of concrete, cement, and bricks [10], many other applications have been investigated, including metal extraction [11,12]. During the last few years, attention has been paid to the study of recovering rare earth elements (REEs) from CFA [13–21].

Rare earth elements are a group of elements consisting of yttrium and 14 other lanthanide elements [22], which are active metals with very similar chemical properties [23,24]. Due to their unique chemical and physical properties, the demand for REEs has expanded in recent years. REEs are essential and necessary components of high-tech products and significant defense applications, such as electronics, space technology, nuclear energy lasers, guidance systems, and radar and sonar systems [25–28]. Considering their crucial role

in modern economics, many countries have listed almost all the 15 rare earth elements as critical raw elements or materials, such as the US, Australia, Japan, and the European Union [29]. However, these elements are becoming critical and scarce as their demand has multiplied due to their wide application. In addition, supplies of these critical elements are controlled by a limited number of sources or countries. For example, the USGS (the United States Geological Survey) Mineral Commodity Summaries 2020 shows that China contributed about 63% of the world's REE mine production in 2019 [30]. As is known to all, China is the largest producer of REEs worldwide. However, at the same time, due to its rapid economic and technological development, China is also the largest consumer around the world. In the year 2015, 60% of the world's REEs consumption was consumed by China, and there is a trend of that increasing year by year. According to estimates, China has consumed 149,000 tons of REE by 2020, up from 98,000 tons in 2015 [31]. Based on the tense relationship between supply and demand, exploring new alternative sources of REEs has become a global hot topic. Researchers from different countries are working on it and have made some progress [32].

Coal is a complex mixture composed of organic and inorganic compounds, and more than 120 kinds of minerals have been identified in coal [33,34]. Three rare earth minerals that always occur as minor mineral constituents in most coal samples are rhabdophane $((Nd, Ce, La)(PO_4) \cdot H_2O)$, monazite $((Ce, La, Th, Nd)PO_4)$, and xenotime $((Y, Er)PO_4)$ [35]. Due to its low volatility, REEs are almost always retained in the residue ash after coal combustion, and there is no significant fractionation between bottom ash and fly ash [36,37]. Looking for alternatives to rare earth mines and realizing the recycling of REEs is significant for improving the current status of rare earth resources and national strategic security. For this purpose, the US Department of Energy (DOE) initiated a rare earth element program to recover rare earth elements from coal and coal byproducts [38].

Understanding the distribution and speciation of rare earth elements in coal fly ash is very important to developing economically viable and environmentally friendly technologies for recovering REEs from CFA. For this purpose, we investigated an industrial CFA from a coal-fired power plant in Guizhou province of southwest China. This work aims to reveal how REEs exist in coal fly ash and put forward ways to improve the recovery rate of REEs. This study will contribute to existing knowledge and also provide useful information, vital for the development of technologies for REE recovery from CFA for future research.

## 2. Materials and Methods

### 2.1. Sample Collection

Samples of pristine CFA were collected from the bottom of the electrostatic precipitators (ESPs) of the Qianxi coal-fired power plant, located in Guizhou province, southwest China, during the year 2017. Four pulverized coal boilers are installed in this power plant, two of which are the HG-1025/17.3-WM18 type, and the other two are the B&WB-1025/17.4-M type, totally equipped with 1200 MW power. The temperature inside the furnace zone is about 1500 °C (>2732 °F). The Qianxi power plant uses local anthracite coal as the feed fuel. The samples were kept dry and tightly sealed.

### 2.2. Sample Preparation and Data Analysis

#### 2.2.1. Sample Preparation

Before analysis and tests, samples were oven-dried at 55 °C for 6 h and stored in a dry and shady place. Each of the dried samples was homogenized well for the next tests and experiments, such as particle size analysis, X-ray diffraction (XRD, Malvern Panalytical, Malvern, Worcestershire, UK) analysis, electron probe microanalysis (EPMA, Shimadzu, Kyoto, Japan) analysis, scanning electron microscopy (SEM, Hitachi, Tokyo, Japan), a grinding experiment, and an acid-leaching experiment. The samples used for XRD analysis were ground to a uniform powder that passed through a 200-mesh (75 µm) sieve. Samples for the SEM and EPMA analysis were mounted as powder grain mounts on

10 mm in diameter metal stubs backed with conductive carbon tape. CFA particles were also mounted in epoxy, and the epoxy mount surface containing the coal fly ash particles was ground sequentially with 180, 400, 800, and 1200 grit metal grinding wheels, and a final polish was applied using 0.05 μm alumina polishing solution. The grain and polished mounted samples were always coated with carbon or platinum before experiments.

### 2.2.2. Data Analysis

The REEs were defined as the whole rare earth elements, which consists of La, Ce, Pr, Nd, Sm, Eu, Gd, Tb, Dy, Ho, Er, Tm, Yb, Lu, and Y. Traditionally, the REEs are divided into LREEs and HREEs based on their atomic weight [39]; the full name of the LREEs is light rare earth elements, which include La, Ce, Pr, Nd, and Sm, while the full name of HREEs is heavy rare earth elements, which consists of Eu, Gd, Tb, Dy, Ho, Er, Tm, Yb, Lu, and Y. The full name of REO is rare earth oxide. Seredin and Dai [4] classified the REEs into three economic clusters based on their relative demand in industry: Critical (Nd, Eu, Tb, Dy, Y, and Er), Uncritical (La, Pr, Sm, and Gd), and Excessive (Ce, Ho, Tm, Yb, and Lu). $C_{outl}$ is the outlook coefficient of the REE composition, which is the ratio of Critical to Excessive.

### 2.3. Characterization

### 2.3.1. Size Analysis

The particle size distribution of this CFA sample was analyzed using the Malvern Instruments Mastersizer 2000 particle size analyzer (Malvern Panalytical, Malvern, Worcestershire, UK). In order to determine the dependence of the content of rare earth elements on the particle size, according to the GB/T 477-2008 standard [40], the fly ash was separated into five-grain classes by the method of wet sieve analysis with four mesh sizes (74, 38.5, 15, and 5 μm).

### 2.3.2. XRD

The bulk mineralogy of the CFA sample was determined by XRD. Powdered samples were mounted in cavity mounts on an automatic sample changer with a spinner. The diffractograms were obtained using a PANalytical X'pert Plus instrument equipped with a programmable incident beam slit and an X' Celerator detector at the University of Arizona. The X-ray radiation used was Ni-filtered Cu K$\alpha$, $\lambda$ = 1.5418 Å. Measurements were made in the bisecting geometry. The sample was scanned at 45 kV, the diffractograms were recorded in the 2$\theta$ angle range from 5° to 70°, with a 0.02° step, during 2 s, and the crystalline substances or minerals in the CFA were identified using the Panalytical High Score software (Version 3.0, Malvern Panalytical, Malvern, Worcestershire, UK) compared with patterns in the diffraction (PDF) database of ICDD [41].

### 2.3.3. XRF

The major elements in this CFA, including $SiO_2$, $Al_2O_3$, $CaO$, $Fe_2O_3$, $MnO$, $TiO_2$, $SO_3$, $K_2O$, $Na_2O$, $MgO$, and $P_2O_5$, were determined by X-ray fluorescence (XRF, Ametek, Berwyn, Pennsylvania, USA) spectrometry—Spectro XEPOS.

### 2.3.4. ICP-MS

However, inductively coupled plasma–mass spectrometry (ICP-MS, Thermo Fisher Scientific, Waltham, Massachusetts, USA) has become one of the most powerful and reliable methods for determining lanthanides. It is a highly sensitive technique for determining ultra-trace REEs in soil, sediment, seawater, and coal fly ash [42]. The concentration of REEs and other trace elements in this sample was measured by ICP-MS (ThermoFisher iCAP-Qc, Thermo Fisher Scientific, Waltham, Massachusetts, USA). The main working parameters of the ICP-MS is listed in Table 1 below.

**Table 1.** Typical operating conditions for the iCAP-Qc used throughout the study.

| ICP-MS Parameter | Value |
|---|---|
| RF power | 1550 w |
| S/C temperature | 2.5 °C |
| Sample Depth | 5 mm |
| Cool gas flow | 14.01 L/min |
| Auxiliary gas flow | 0.78741 L/min |
| Nebulizer gas flow | 0.9941 L/min |
| Pump | 40 r/min |
| Dell time (S) | 0.02 s |
| Number of sweeps | 50/e |

For ICP-MS analysis, the 0.1 g CFA samples were accurately weighted into a 100 mL Teflon digestion vessel, in which 5 mL of HF, 4 mL of $HNO_3$, and 2 mL of $H_2SO_4$ (*v/v*, 50%) was added. After one hour, the sample was placed into a microwave digestion system (CEM MARS 6) and heated with a digestion program, as listed in Table 2 below. After digestion, the sample solution was filtered (nylon syringe filter with a 0.22 μm pore size) into a 50 mL volumetric flask and the filtrate diluted to volume with water. Fifteen rare earth elements mixed standards (GSB 04-1789-2004) were used for calibration of the trace element concentrations. To try and reduce possible interference during the ICP-MS analysis, the addition of a few drops of concentrated sulfuric acid ($H_2SO_4$) was used to remove the excess of HF to avoid damage to the quartz and glass parts of the ICP-MS equipment.

**Table 2.** Microwave program for the CFA sample digestion.

| Step | Microwave Power/W | Temperature/°C | Time/min |
|---|---|---|---|
| 1 | 1000 | 60 | 12 |
| 2 | 1000 | 125 | 20 |
| 3 | 1000 | 160 | 8 |
| 4 | 1000 | 240 | 15 + 60 (maintain) |

2.3.5. SEM-EDS and EPMA-WDS

The morphology, microstructure, and chemical content of the fly ash particles were carried out with a Hitachi S-3400N variable pressure SEM equipped with an Oxford Inca Energy 350 X-act energy dispersive X-ray analyzer (EDS, Oxford Instruments, Oxfordshire, UK) (analyses were executed on the Imaging Cores on the University of Arizona). A backscattered electron detector was used at 30 kV so that the elemental contrast could be used to locate the REE-containing carriers of interest more easily. An electron probe microanalyzer (EPMA-8500G, Shimadzu, Kyoto, Japan) was also conducted to study the REE carriers in this study. An X-ray diffraction (WDS) spectrometer (Shimadzu, Kyoto, Japan) was used to quantitatively analyze the elemental composition, and the analysis was performed with a current of 45 nA and an accelerating voltage of 20 kV, while the map dwell time was 250 ms.

2.3.6. Grinding–Leaching Experiments

The coal fly ash sample was mechanically ground by a laboratory stirring ball mill (QHJM-1) with different grinding times (20, 40, and 60 min) to destroy the particle structure of the glassy components of the CFA samples. In total, a 50 g raw coal fly ash sample and 100 mL deionized water were added to the ball mill for the different grinding time experiments. The stirring shaft working speed was 650 rpm and the grinding medium material was zirconia balls with a diameter of 5 mm. After grinding, the sample was filtered and dried for the leaching experiments on REEs extraction. Sample leaching was conducted by heating 1 g of coal fly ash and 7 mL of aqua regia for 2.5 h in a 25 mL Teflon beaker on a constant temperature heating plate at 75 °C [43]. The leachate was then separated

by filtration using a nylon syringe filter with a 0.22 μm pore size, and the residue was washed four times with deionized water. In the end, the leachate and the washed water were diluted to 50 mL and sent for ICP-MS analysis to determine the REEs' concentration.

To evaluate the enhanced function of pre-treatment on the leaching process, the leaching efficiency was calculated according to Equation (1) [44]:

$$\alpha = \frac{VC_2}{MC_1} \times 100\% \tag{1}$$

where $V$ is the volume of leachate, in mL; $M$ is the mass of coal fly ash sample, in g; $C_1$ is the element content in the coal fly ash sample, in μg/g; and $C_2$ is the element concentration in the leachate, in μg/mL.

## 3. Results and Discussion

### 3.1. Coal Fly Ash Properties

Figure 1 shows the particle size distribution curve of the CFA within the scope of particle diameter between 0.8 and 360 μm. D10, D50, and D90 of the examined coal fly ash are 6.63 μm, 34.29 μm, and 108.65 μm, respectively (D10, D50, and D90—characteristic grain diameters below which 10%, 50%, and 90% of the analyzed material is found, respectively). The particle size of the coal fly ash is relatively fine. XRD studies (Figure 2) identified that the major mineral composition in the coal fly ash were quartz and mullite. The XRD pattern humped from 15° to 40° (2θ), which indicated the presence of amorphous material, including amorphous glass and unburned charcoal grains.

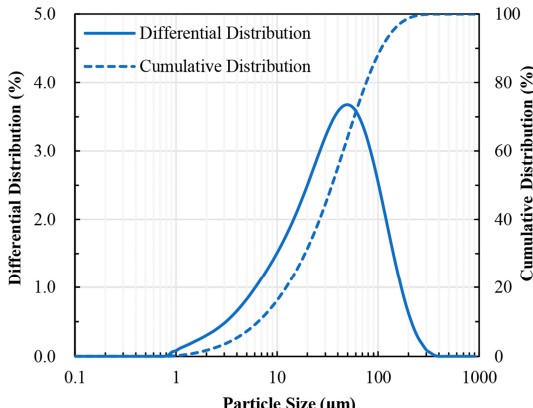

**Figure 1.** Particle size distribution in the CFA sample from the Qianxi power plant in Guizhou.

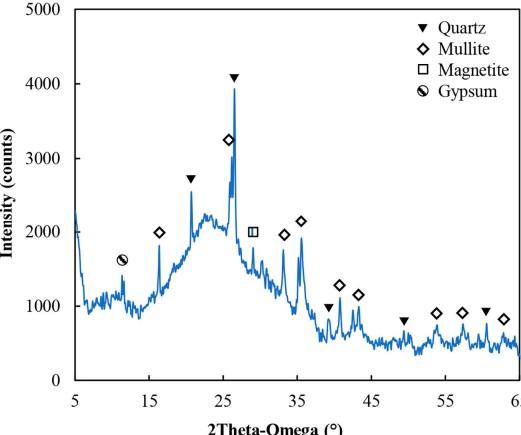

**Figure 2.** XRD patterns of the CFA sample from the Qianxi power plant in Guizhou.

### 3.2. Chemical Components and REEs Content of CFA

The coal fly ash sample comprises cenospheres, solid iron spheres, Al/Si/Ca slag pellets, aluminosilicate glass, and solid Ca oxide particles. As shown in Table 3, the dominant chemical components of the CFA sample were $SiO_2$, and $Al_2O_3$, which account for approximately 65% of the whole sample. The share of $SiO_2$ in the analyzed sample was 44.5%, while $Al_2O_3$ was 20.8%. The other chemical compositions in percent by mass are $Fe_2O_3$ (12.11 wt%), $TiO_2$ (3.56 wt%), CaO (2.72 wt%), $Na_2O$ (2.08 wt%), $K_2O$ (1.80 wt%), $SO_3$ (1.15 wt%), and MgO (1.06 wt%). The remaining chemical components, $P_2O_5$ and $Mn_3O_4$, are in smaller amounts, with the proportions generally not exceeding 0.50 wt%. Loss on ignition (LOI) of fly ash was determined by heating the sample at 1050 °C for at least one hour. The test shows that the LOI of this CFA is 9.68%.

**Table 3.** The main chemical components' content in the CFA sample from the Qianxi power plant in Guizhou (in wt%) [a].

| Elements | $Na_2O$ | MgO | $Al_2O_3$ | $SiO_2$ | $P_2O_5$ | $SO_3$ | $K_2O$ | CaO | $TiO_2$ | $Mn_3O_4$ | $Fe_2O_3$ | LOI [b] |
|---|---|---|---|---|---|---|---|---|---|---|---|---|
| Content | 2.08 | 1.06 | 20.80 | 44.50 | 0.38 | 1.15 | 1.80 | 2.72 | 3.56 | 0.15 | 12.11 | 9.68 |

[a] Quantified by XRF analysis. [b] Loss of ignition based on ASTM standard D3174.

It can be stated that the coal fly ash is low-calcium fly ash (CaO mass fraction < 10%). According to the ASTM C618-15(2015) standard, coal fly ash is divided into two types based on the CaO content, Class C (CaO > 20%) and Class F (CaO < 20%), and the coal fly ash in this study is typically a Class F ash. The iron content in the coal fly ash is relatively high. $Fe_2O_3$ accounts for 12.55%. Many of these irons will exist as dispersed iron oxide particles, forming spinels related to magnetite ($Fe_3O_4$), maghemite ($\gamma$-$Fe_2O_3$), and hematite ($Fe_2O_3$). The remaining iron may exist in the glass phase as well as mullite or other crystals. It also shows that the magnetic field can separate some particles in the coal fly ash from other non-magnetic particles.

Table 4 presents the concentration of REEs measured by ICP-MS for CFA from the Qianxi power plant. It is visible that the content of REEs in the CFA sample is highly variable compared with the Clarke value in the earth's crust. Bulk elemental analysis of the samples by ICP-MS indicates that the total content of REEs in the coal fly ash sample is 630.51 ppm, which is approximately four times higher than that in the earth's crust and higher than the average for the world's hard coal ashes of 403.50 ppm. The REO content calculated as oxides in this sample is 756.61 ppm, much higher than the average for hard coal ash from global deposits of 484.2 ppm [45,46]. Light rare earth elements (LREEs) have the highest content, approximately 77.85%. According to the different threshold values, the REEs' concentration often evaluates the economic viability of recovering REEs from CFA. The US DOE recommends coal ash with REEs higher than 300 ppm as the candidate resource for recovery. However, Seredin and Dai suggested that critical REEs higher than 30% and the prospect coefficient ($C_{outl}$) greater than 0.7 allowed economic recovery of REEs. It can be seen that the Guizhou coal fly ash we selected has specific economic feasibility for REEs recovery.

**Table 4.** Contents of the rare earth elements in coal fly ash from the Qianxi power plant in Guizhou compared with other samples (ppm).

| Element | Content | | | |
|---|---|---|---|---|
| | Coal Fly Ash Sample (This Study) | World Coal Ashes [47] | World Coals [47] | Earth's Crust [48] |
| Y | 75.94 | 51.00 | 8.40 | 28.10 |
| La | 113.07 | 69.00 | 11.00 | 18.30 |
| Ce | 228.13 | 130.00 | 23.00 | 46.10 |
| Pr | 26.79 | 20.00 | 3.50 | 5.53 |

**Table 4.** *Cont.*

| Element | Content | | | |
|---|---|---|---|---|
| | Coal Fly Ash Sample (This Study) | World Coal Ashes [47] | World Coals [47] | Earth's Crust [48] |
| Nd | 103.16 | 67.00 | 12.00 | 23.9 |
| Sm | 19.70 | 13.00 | 2.00 | 6.47 |
| Eu | 3.61 | 2.50 | 0.47 | 1.06 |
| Gd | 20.44 | 16.00 | 2.70 | 6.36 |
| Tb | 2.75 | 2.10 | 0.32 | 0.91 |
| Dy | 15.69 | 14.00 | 2.10 | 4.47 |
| Ho | 2.90 | 4.00 | 0.54 | 1.15 |
| Er | 8.45 | 5.50 | 0.93 | 2.47 |
| Tm | 1.12 | 2.00 | 0.31 | 0.20 |
| Yb | 7.73 | 6.20 | 1.00 | 2.66 |
| Lu | 1.03 | 1.20 | 0.20 | 0.75 |
| LREEs | 490.85 | 299.00 | 51.50 | 100.30 |
| HREEs | 139.66 | 104.50 | 16.97 | 48.13 |
| REEs | 630.51 | 403.50 | 68.47 | 148.43 |
| REO | 756.61 | 484.20 | 82.164 | 178.116 |
| Critical | 209.60 | 142.10 | 24.22 | 60.91 |
| Uncritical | 180.01 | 118.00 | 19.20 | 36.66 |
| Excessive | 240.91 | 143.40 | 25.05 | 50.86 |
| Critical (%) | 33.24 | 35.22 | 35.37 | 41.04 |
| $C_{outl}$ | 0.87 | 0.99 | 0.97 | 1.20 |

In this study, the CFA was separated using the wet sieving method to study the REEs' distribution in different size fractions, namely, +74 μm, −74 + 38.5 μm, −38.5 + 15 μm, −15 + 5 μm, and −5 μm. The weight distribution, REEs distribution, and REEs concentration of the different size fractions are presented in Figure 3. It can be concluded from this figure that, in general, REEs are distributed in each size fraction but more distributed in the fine particles. In total, 53.25% of the rare earth elements are distributed in the minus 15 μm size fraction products, yielding 43.22%. The results showed that the REEs concentration slightly increased from 448.62 ppm to 913.28 ppm with a decrease in particle size, and the highest REEs concentration was measured at the −5 μm size fraction, which has the highest REEs enrichment factor of 1.45. The content of some volatilized elements, such as sodium and potassium, increases with the decrease in the particle size because the volatilized elements could condense on the surface of the fine particles [43], which usually have a larger specific surface area than coarse particles. Although the rare earth elements are not volatilized elements, however, they are generally enriched in the low-content accessory minerals, such as monazite and xenotime [4,49], in coal before combustion. These rare earth element-enriched mineral particles will decompose and break into finer particles during combustion. Moreover, during the formation of coal ash, part of these fine REE carrier particles will be encapsulated by larger glass bodies during cooling. That is why there is also a specific content of REEs in the large size fraction.

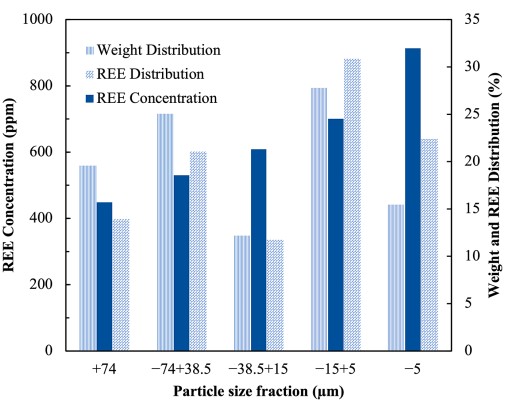

**Figure 3.** Concentration and distribution of REEs in the CFA sample by size fraction.

From the above study, it can be seen that the content of REEs in fly ash is very low; it is impossible to detect minerals containing rare earth elements by XRD test directly, but it is still possible to observe REE-rich minerals and study the speciation of REEs in fly ash utilizing scanning electron microscopy and electron probe microscopy. The REE-bearing mineral phases usually appear to be very bright in the ash pellets in the backscattered mode of SEM-EDS, since the brightness of the backscattered imaging depended on the average atomic number of the material [50]. So, EPMA-WDS is similar in scope to the SEM method. Theoretically, if there are rare earth particles dispersed in fly ash, they can be identified using SEM-EDS.

### 3.3. The Speciation of REEs in Coal Fly Ash

SEM and EPMA analysis provide morphological and elemental data to identify the REE carriers and REE-bearing mineral phases in coal fly ash, which provides vital information about the target REE-bearing mineral phases. In this study, a few REE-bearing particles were identified via EPMA-WDS and SEM-EDS in the coal fly ash sample (Figures 4–6). An irregularly elongated-shaped amorphous carrier and an irregularly shaped REE mineral-bastnasite were overserved by SEM-EDS. Panel a) in Figure 4 shows an irregularly shaped amorphous particle that may have been due to rapid cooling, which is composed of Al/Si/Ca and contains iron oxide and rare earth elements (Nd/Sm/Gd). Panel b) in Figure 4 shows an irregularly shaped crystalline REE mineral (bastnasite) particle, in which the REEs (La/Ce/Pr/Nd) make up about 45%. The REE particle in Figure 5 reveals that the REEs in this particle can be dispersed throughout the aluminosilicate glass phase.

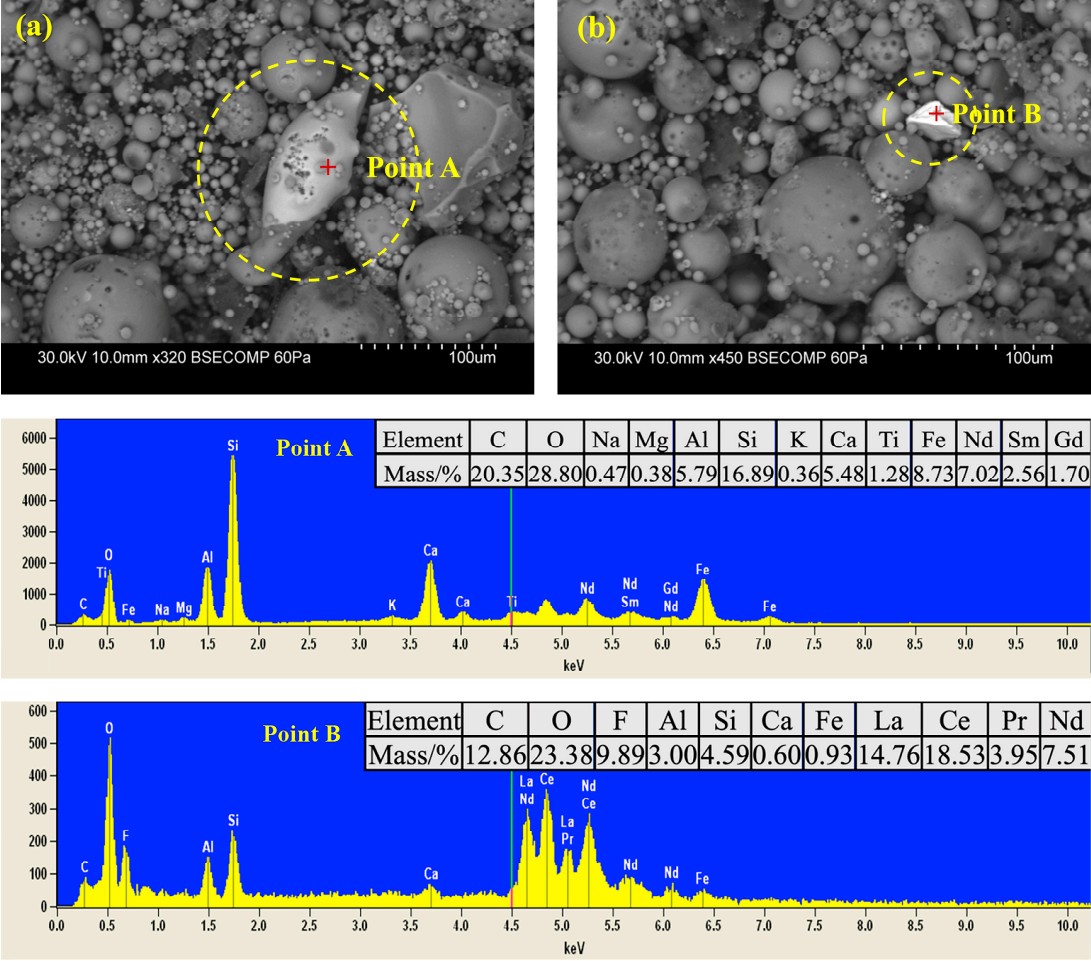

**Figure 4.** BSE image with element spectra of (**a**) an irregularly elongated-shaped amorphous carrier and (**b**) an irregularly shaped REE mineral—bastnasite.

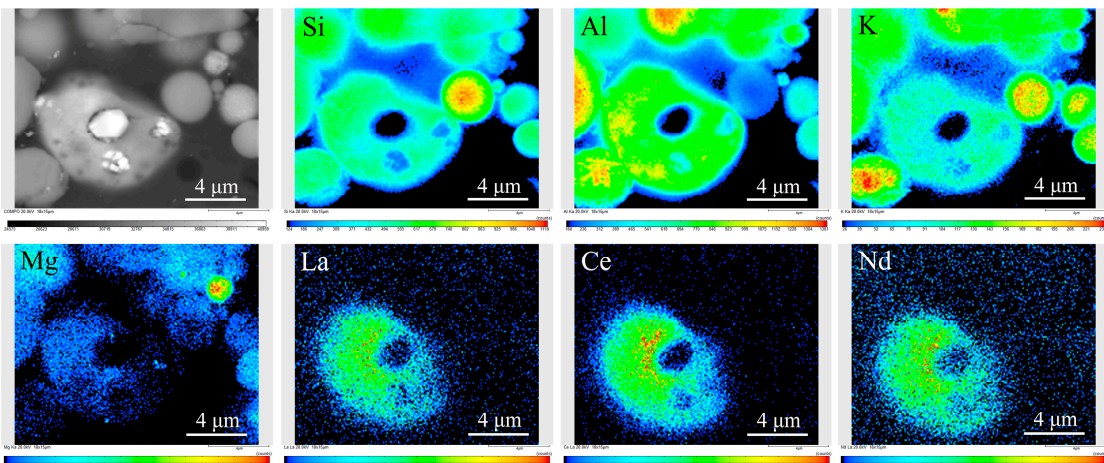

**Figure 5.** EPMA-WDS quantitative elemental mapping of the coal fly ash sample. The scale bar is 4 μm. Analysis conditions: 20 kV, 45 nA, and a map dwell time of 250 ms.

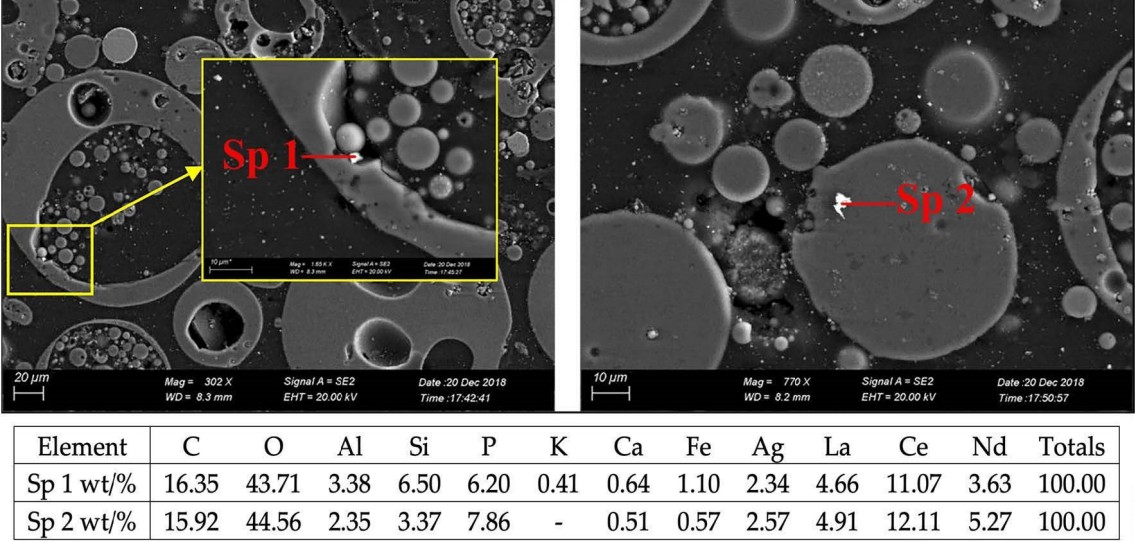

| Element | C | O | Al | Si | P | K | Ca | Fe | Ag | La | Ce | Nd | Totals |
|---------|------|-------|------|------|------|------|------|------|------|------|-------|------|--------|
| Sp 1 wt/% | 16.35 | 43.71 | 3.38 | 6.50 | 6.20 | 0.41 | 0.64 | 1.10 | 2.34 | 4.66 | 11.07 | 3.63 | 100.00 |
| Sp 2 wt/% | 15.92 | 44.56 | 2.35 | 3.37 | 7.86 | - | 0.51 | 0.57 | 2.57 | 4.91 | 12.11 | 5.27 | 100.00 |

**Figure 6.** SEM images of the irregularly shaped REE-bearing minerals encapsulated within glassy particles in the coal fly ash sample.

Studies have shown that there are rare earth minerals such as monazite, xenotime, and apatite in coal [33]. The rare earth elements in coal are mainly concentrated in these rare earth minerals dispersed in coal, while their melting points are often very high (the melting point of natural monazite ranges from 1916–2072 °C [51]); so, these rare earth minerals will remain in the fly ash in the form of irregular crystal minerals after combustion. Rare earth particles tend to be irregular, and spherical shapes are rarely observed. Due to the relatively low content of REEs, it is often difficult to find rare earth-rich particles in coal fly ash using scanning electron microscopy. Although challenging, some rare earth particles could be found with painstaking search using the SEM-EDS with the backscattered detector.

REE carriers are present in ash samples as individual crystal or amorphous particles in the ash matrix, but it is also possible to exit inside the glass, which is difficult to be observed by SEM and EPMA. Based on this consideration, a fly ash sample of epoxy resin was made to study the internal situation of the fly ash particles. Figure 6 demonstrates that REE carriers could be encapsulated in large grains as micro-particles.

The speciation of REEs in coal fly ash is a key factor affecting the recovery efficiencies of REEs from CFA. Coal combustion is a complex chemical and physical process that includes heterogeneous reactions. Based on the above study of REEs speciation in CFA and

the glass formation mechanism during coal combustion, this study classifies the speciation of REEs in coal fly ash as (1) amorphous glassy particles, with REE minerals or compounds encapsulated inside; (2) amorphous glassy particles, with REEs distributed throughout; and (3) discrete REE minerals or compounds.

### 3.4. Effect of Grinding on Acid Leaching

It can be seen from the above research that during the formation process of fly ash, a considerable part of the rare earth particles was sequestered in the glass body. As seen in Table 5, the leached rare earth elements increased from 144.29 to 263.74 ppm when the wet grinding time increased from 0 to 60 min, and the leaching proportion of heavy rare earth elements increased from 19.14% to 21.69%.

**Table 5.** Contents of the rare earth elements after leaching at different grinding times (ppm).

| Element | No Grinding | Grinding 20 min | Grinding 40 min | Grinding 60 min |
|---|---|---|---|---|
| Y | 15.22 | 24.87 | 27.62 | 30.89 |
| La | 33.33 | 37.33 | 41.86 | 47.02 |
| Ce | 51.16 | 72.96 | 80.50 | 89.53 |
| Pr | 6.59 | 10.46 | 12.02 | 13.59 |
| Nd | 21.43 | 35.92 | 41.48 | 46.61 |
| Sm | 4.17 | 7.75 | 8.43 | 9.80 |
| Eu | 0.81 | 1.40 | 1.60 | 1.77 |
| Gd | 4.26 | 7.29 | 8.23 | 8.94 |
| Tb | 0.62 | 0.97 | 1.18 | 1.26 |
| Dy | 3.16 | 4.97 | 5.49 | 6.41 |
| Ho | 0.56 | 0.86 | 1.01 | 1.14 |
| Er | 1.55 | 2.47 | 2.85 | 3.29 |
| Tm | 0.19 | 0.30 | 0.37 | 0.43 |
| Yb | 1.07 | 1.99 | 2.28 | 2.63 |
| Lu | 0.17 | 0.32 | 0.39 | 0.44 |
| REEs | 144.29 | 209.88 | 235.30 | 263.74 |
| LREEs | 116.67 | 164.42 | 184.28 | 206.55 |
| HREEs | 27.62 | 45.45 | 51.02 | 57.20 |
| Leaching proportion (LREEs) [a] | 80.86% | 78.34% | 78.32% | 78.31% |
| Leaching proportion (HREEs) [b] | 19.14% | 21.65% | 21.68% | 21.69% |

[a]. Leaching proportion (LREEs) $= \frac{\Sigma LREEs}{\Sigma REEs} \times 100\%$; [b]. Leaching proportion (HREEs) $= \frac{\Sigma HREEs}{\Sigma REEs} \times 100\%$.

The original D90 before grinding was as high as 106.65 μm, while after wet grinding with a ball mill for 20 mins, the D90 dropped sharply to 15.54 μm (Figure 7). The particle size of 90% of the fly ash was below 3.5 μm after 1 h of grinding, the spherical glass particles were broken, and the internal phases got to be released (Figure 8). Through acid leaching with aqua regia on the ground fly ash samples, with the increase in grinding time, the REEs leached by aqua regia acid also increased continuously. Before grinding, 23.49% of the total REEs can be leached. After one hour of grinding, the rare earth elements leached by aqua regia acid accounted for 41.68% of the total REEs (Figure 7). It shows that a considerable part of the REEs was encapsulated in the glass body of fly ash. Since the main phase glass body of the fly ash is a dense silica-alumina glass body, it is difficult for acids such as HCl, HNO$_3$, and H$_2$SO$_4$ to dissolve because it is sealed. The rare earth minerals inside the particle cannot react and dissolve in contact with the acid. After grinding, this part of the rare earth element minerals can be liberated and released from the glass body so that it can be extracted by aqua regia leaching. However, approximately 58% of the rare earth elements have not been extracted by aqua regia, possibly because the dissociation is insufficient. Another reason is that some rare earth minerals are decomposed into rare earth oxides at high temperatures and participate in the composition of the glass body. It is

the chemical composition of the amorphous glass body, and the REEs in this part are also challenging to be leached by aqua regia.

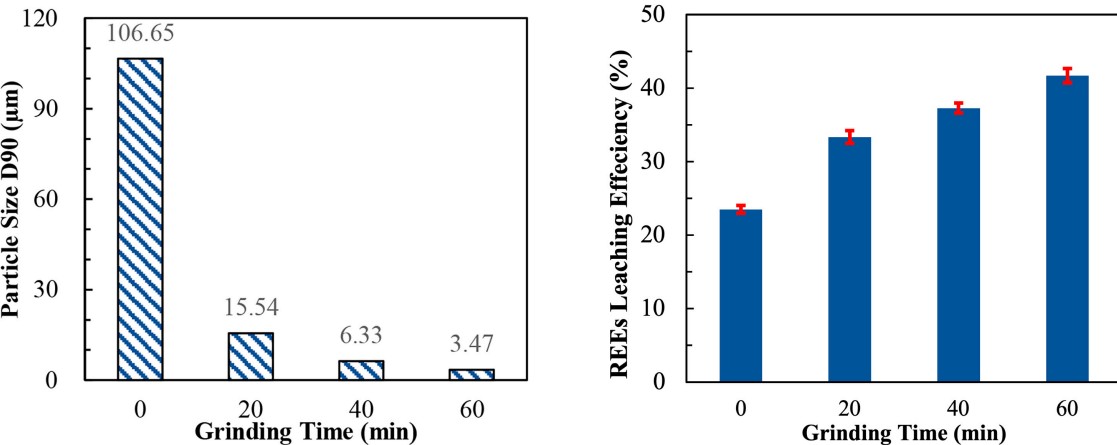

**Figure 7.** The effect of grinding time on the particle size and REE leaching efficiency of CFA.

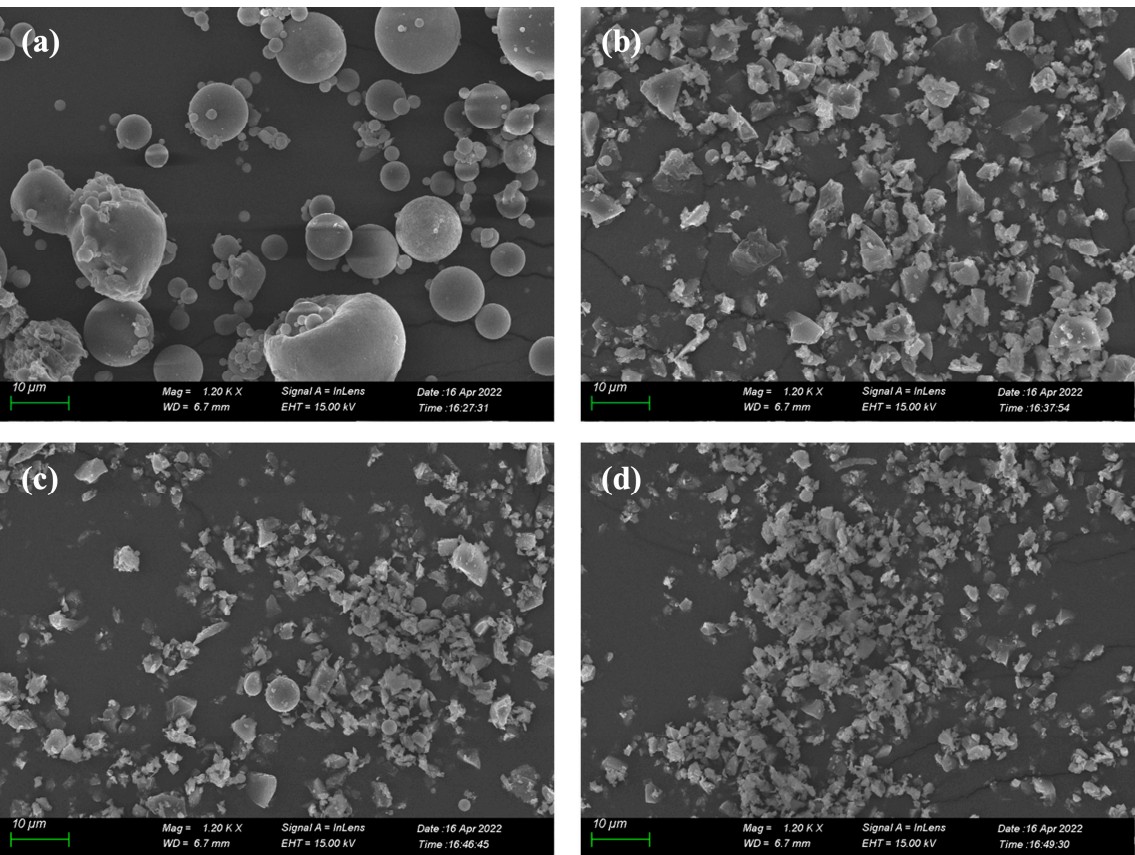

**Figure 8.** SEM microstructures of the CFA sample after grinding for 0 to 60 min (**a**–**d**).

Research has shown that presenting rare earth oxides into glass causes variation in the structure species $Q^n$ with different non-bridging oxygen (NBO) [52], increasing NBO and decreasing the glass network connectivity. The addition of rare earth oxides can improve the chemical durability of glass because of the depressing and hindering effect of rare earth oxides on the moving of alkali cations and exchange reaction between an alkali cation and $H^+$ ($H_3O^+$). Therefore, when rare earth oxides are involved in forming the glass body (such as the particle in Panel a) in Figure 4 and the particle in Figure 5), it is difficult to extract this part of the REEs by acid leaching. Wet grinding can improve a specific recovery rate,

but this increase is limited. More rare earth elements are trapped in the structure of the glass body and become part of the structure, which is difficult to extract by acid leaching.

## 4. Conclusions

In this study, the REE content of CFA from the Qianxi power plant was studied, which is valuable for understanding REE occurrence in CFA and the potential for recovery. The results showed that the content of REEs in the CFA sample was 630.51 ppm. It can be revealed through the microanalysis that the REEs in CFA is not uniformly dispersed throughout the whole fly ash but only enriched in a few particles. The amorphous glassy particle with REE minerals or compounds encapsulated inside, the amorphous glassy particle with REEs distributed throughout, and discrete REE mineral particles were observed directly by SEM and EPMA. The wet grinding-enhanced leaching experiments revealed that a part of the rare earth particles was encapsulated within the glass body, and these rare earth particles can be liberated and released to a certain extent by grinding. The results suggested that wet grinding would increase the aqua regia leaching recovery of REEs from 23.49% to 41.68%. The above research results are a basis for developing an economically viable and environmentally benign technology for REEs recovery from CFA, as a promising alternative source.

**Author Contributions:** L.W.: data processing and, writing—original draft; L.M.: supervision and funding acquisition; G.H., J.L., H.X.: writing—review and editing. All authors have read and agreed to the published version of the manuscript.

**Funding:** This work was supported by the National Key R&D Program of China (Grant NO. 2021YFC29026), National Natural Science Foundation of China (Grant NO. 52004294), and the Fundamental Research Funds for the Central Universities (Grant NO. 2021YQHH01).

**Acknowledgments:** Many thanks are given to the academic editors and reviewers for their careful and valuable comments, which improved the quality of this manuscript.

**Conflicts of Interest:** We declare that we have no financial and personal relationships with other people or organizations that can inappropriately influence our work, there is no professional or other personal interest of any nature or kind in any product, service, and/or company that could be construed as influencing the position presented in, or the review of, the manuscript entitled.

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
