# Peer review of "Distribution and Speciation of Rare Earth Elements in Coal Fly Ash from the Qianxi Power Plant, Guizhou Province, Southwest China"

_minerals, doi:10.3390/min12091089_

Round 1

Reviewer 1 Report

Dear Authors,

The following are my comments:

Line 8-10: the sentence is too wordy and can be made simple for clarity.

Line 49-50: Include the full name of USGS.

Line 53-54: In the year of 2015, 60 percent of world consumption was consumed by China. Please be specific about what was consumed by China. Also, “In the year of 2015” – “In the year 2015”

Line 57: the word “treasure hunt” can be replaced.

Line 60: change “As is known to all, coal is a complex mixture composed of organic and inorganic” to “Coal is a complex mixture composed of organic and inorganic” because “coal” might not be known to all.

Materials and methods

Please reorganise your materials and method section as at the moment it is hard to follow. Please consider subdividing the section into the following:

1.       Sample collection

2.       Sample preparation and data analysis – this should include in clear terms the methods used and how it was applied. Provide the solid/liquid ratio or amount as well as the formulas used for the different data analyses like critical, uncritical, critical percentage, coefficient (Coutl), etc. Please include the different variables considered in the study during sample preparations particle size, grinding, acid leaching, etc.  

3.       Characterization

Please the sections should be kept concise and with a clear procedure considering all the variables in the study.

Results and discussion

Line 139-142 “Compared with the development and utilization of natural rare earth ore resources, as a new type of rare earth resource, fly ash has the advantage of reducing the high cost and high energy consumption operations such as mining, crushing, and grinding”

Please clearly distinguish this claim from other published manuscripts. A table might be useful in this regard, which will clearly show the difference.

Please reconsider this sentence to be moved “The coal fly ash sample comprises cenospheres, solid iron spheres, Al/Si/Ca slag pellets, aluminosilicate glass, and solid Ca oxide particles”. This can be related to the elemental composition of the samples.

Line 147: “XRD pattern baseline was raised from 15° to 40° (2θ)” Please change “baseline was raised” to “hump (from 15° to 40° (2θ))”

Line 149-151: “Because the content of REEs in fly ash is too low, it is impossible to detect minerals containing rare earth elements by XRD test directly, but it is still possible to observe REEs-rich minerals in fly ash utilizing scanning electron microscopy and electron probe microscopy”.

This claim cannot be made at this point. I will suggest this be moved to the chemical composition section.

Please it is mandatory to add a critical discussion of your results with state-of-the-art literature after the presentation of the results from the experiments. Some claims have not been substantively backed by pieces of literature (using references) all through the manuscript.

Line 179-180: “The characterization of the coal fly ash suggests that the REEs in the samples is too dispersed to be identified by XRD analysis of a powdered sample” is more or less the same as lines 89-91: “Powerful and reliable methods for determining lanthanides. It is a highly sensitive technique for determining ultra-trace REEs in soil, sediment, seawater, and coal fly ash [28]. The concentration of REEs and other trace elements in this sample was measured by ICP-MS”.

Light rare earth elements “LREY” = LREE?. Also, include the full name of HREE, and REO. Which group of rare earth are made of LREE and HREE in this study based on Table 2.

Please explain how the following was obtained in Table 2: critical, uncritical, and excessive (as mentioned above in my comment – materials and method).

Line 198: Please correct “+74 μm, 74-38.5 μm, 38.5-15 μm, 15-5 μm, and -5 μm” by adding the minus sign as indicated in Figure 3.

3.3 The speciation of REEs in coal fly ash

Please present the experimental results first before other literature. This makes it difficult to differentiate between the study observation and existing literature.

Figures 4 and 6 showed consistent spherical shapes in all the images. What could be the reason? Also, it was not indicated that all the images have a spherical shape in the SEM results discussion.

3.4 Effect of grinding on acid leaching

Line 261-263: “It can be seen from the above research that during the formation process of fly ash, a considerable part of rare earth particles was sequestered in the glass body, which was very difficult to leach by acid”

Please which of the following SEM images represent the acid leaching process, Figure 6 or 8. Figure 8 showed changes in morphology after the acid leaching process.

Line 279-280: “it is difficult for acids such as HCl, HNO3, and H2SO4 to dissolve because it is sealed”.

Is this more of speculation? Since the experimental indicate an acid leaching of HCL/HNO3. Or can this claim be further supported with literature?

Please reorganise the manuscript as at the moment it is hard to read without clear logic and more speculations with no backing of your results with state-of-the-art literature.

Please have the manuscript checked by a professional editor. 

Reviewer 2 Report

This study on REE content of CFA sourced from coal-fired power plants in China. Though not really novel, it is original and relevant. However, significant changes need to be made, especially in the methodology section. For all of the instruments/techniques (XRF, SEM-EDS, EPMA, ICP-MS, WET GRINDING...) reported to have been used in this study, details of the experimental design and instrument settings used are grossly lacking. Also, references to the used of established method(s) are lacking.

I have left clear comments within the attached reviewed manuscript which need to be addressed to improve the quality of the research.

Thanks.

Reviewer 3 Report

The authors have presented on the distribution and speciation of rare earth elements in coal fly ash from the Qianxi Power Plant. Though experimental work has been well described and the results have been presented where key conclusions have been drawn. The article could serve as a good reference for stakeholders in the industry especially where repurposing of coal fly ash is of interest. There some minor issues which need to be addressed by the authors before final publication.

1. the authors need to be commended for the excellent introduction provided.

2. Referencing of some statements should be improved some examples and suggested references have been provided as follows;

A. Line 40:"which are active metals with similar chemical properties" could be referenced with 

(i) Jordens, A., Cheng, Y.P. and Waters, K.E., 2013. A review of the beneficiation of rare earth element bearing minerals. Minerals Engineering41, pp.97-114.

(ii) Kim, C.J., Yoon, H.S., Chung, K.W., Lee, J.Y., Kim, S.D., Shin, S.M., Lee, S.J., Joe, A.R., Lee, S.I., Yoo, S.J. and Kim, S.H., 2014. Leaching kinetics of lanthanum in sulfuric acid from rare earth element (REE) slag. Hydrometallurgy146, pp.133-137.

B. Line 44: There are numerous reviews which have confirmed this statement, hence authors are encouraged to add at least 3 more  as follows or any others preferred by the authors 

(i) Binnemans, K., Jones, P.T., Müller, T. and Yurramendi, L., 2018. Rare earths and the balance problem: how to deal with changing markets?. Journal of Sustainable Metallurgy4(1), pp.126-146.

(ii) Abaka-Wood, G.B., Ehrig, K., Addai-Mensah, J. and Skinner, W., 2022. Recovery of Rare Earth Elements Minerals from Iron-Oxide-Silicate-Rich Tailings: Research Review. Eng3(2), pp.259-275.

(ii) Balaram, V., 2019. Rare earth elements: A review of applications, occurrence, exploration, analysis, recycling, and environmental impact. Geoscience Frontiers10(4), pp.1285-1303.

C. Line 184-186 needs a supporting reference see 

(i) Abaka-Wood, G.B., Addai-Mensah, J. and Skinner, W., 2022. The concentration of rare earth elements from coal fly ash. Journal of the Southern African Institute of Mining and Metallurgy122(1), pp.21-28.

(ii) Perämäki, S.E., Tiihonen, A.J., and Väisänen, A.O. 2019. Occurrence and recovery potential of rare earth elements in Finnish peat and biomass combustion fly ash. Journal of Geochemical Exploration, vol. 201. pp. 71-78.

3. Line 206-208: The authors should consider showing the results of sodium and potassium to support the statement made. Plot of sodium and potassium could be included in Figure 3. 

4. How were the leaching proportions (%) and efficiencies calculated? Authors should show the formulae or expressions used.

5. How many times were the leaching and grinding tests carried out to ensure the level of confidence of the results. Errors need to be shown to demonstrate that.  

Round 2

Reviewer 2 Report

The article has been significantly improved.
